# Human β-Defensin 2 (HBD-2) Displays Oncolytic Activity but Does Not Affect Tumour Cell Migration

**DOI:** 10.3390/biom12020264

**Published:** 2022-02-06

**Authors:** Guneet K. Bindra, Scott A. Williams, Fung T. Lay, Amy A. Baxter, Ivan K. H. Poon, Mark D. Hulett, Thanh Kha Phan

**Affiliations:** Department of Biochemistry and Genetics, La Trobe Institute for Molecular Science, La Trobe University, Melbourne, VIC 3086, Australia; g.bindra@latrobe.edu.au (G.K.B.); scott.williams@latrobe.edu.au (S.A.W.); f.lay@latrobe.edu.au (F.T.L.); a.baxter@latrobe.edu.au (A.A.B.); i.poon@latrobe.edu.au (I.K.H.P.)

**Keywords:** defensin, host defence peptide, tumour, cell death, cell migration

## Abstract

Defensins form an integral part of the cationic host defence peptide (HDP) family, a key component of innate immunity. Apart from their antimicrobial and immunomodulatory activities, many HDPs exert multifaceted effects on tumour cells, notably direct oncolysis and/or inhibition of tumour cell migration. Therefore, HDPs have been explored as promising anticancer therapeutics. Human β-defensin 2 (HBD-2) represents a prominent member of human HDPs, being well-characterised for its potent pathogen-killing, wound-healing, cytokine-inducing and leukocyte-chemoattracting functions. However, its anticancer effects remain largely unknown. Recently, we demonstrated that HBD-2 binds strongly to phosphatidylinositol-4,5-bisphosphate (PI(4,5)P_2_), a key mediator of defensin-induced cell death and an instructional messenger during cell migration. Hence, in this study, we sought to investigate the lytic and anti-migratory effects of HBD-2 on tumour cells. Using various cell biological assays and confocal microscopy, we showed that HBD-2 killed tumour cells via acute lytic cell death rather than apoptosis. In addition, our data suggested that, despite the reported PI(4,5)P_2_ interaction, HBD-2 does not affect cytoskeletal-dependent tumour cell migration. Together, our findings provide further insights into defensin biology and informs future defensin-based drug development.

## 1. Introduction

Defensins, a group of cysteine-containing, β-sheet-rich, cationic HDPs, are important contributors to innate immunity, providing crucial infection-combating mechanisms [1,2]. Despite the varying primary sequence, the tertiary structure of defensins remains relatively similar across different kingdoms, suggesting an evolutionarily conserved mechanism of action [3]. Initially discovered for their potent antimicrobial activity, defensins have recently gained increased interest due to their functional complexity, among which are tumour-suppressing effects [4,5,6]. Defensins have, therefore, emerged as a potential new class of multifaceted therapeutic agents [7]. 

Several plant and human defensins such as *Nicotiana alata* defensin 1, NaD1, *Nicotiana occidentalis* defensin, NoD173, tomato defensin TPP3 and human β-defensin HBD-3, selectively target tumour cell membranes and rapidly induce the formation of large, irreversible membrane blebs followed by cell lysis. The defensin-induced oncolysis is mediated by their interaction with membrane phosphoinositides, particularly phosphatidylinositol-4,5-bisphosphate (PI(4,5)P_2_) through a conserved cluster of positively-charged amino acids [4,8,9,10]. In contrast, human neutrophil peptide 1 (HNP-1, an α-defensin) and human β-defensin HBD-1 induce apoptosis, a form of non-lytic programmed cell death, in prostate adenocarcinoma and bladder cancer cells, respectively [11,12]. In vivo studies have shown that NoD173 is effective in arresting mouse melanoma growth in a xenograft tumour model [4]. In addition, defensins can also prevent tumour cell migration via the disruption of cytoskeleton dynamics. For example, PvD1 defensin (from common bean *Phaseolus vulgaris*) effectively disturbs the cytoskeleton of breast cancer cells, thus perturbing cell-to-cell adhesion and tumour cell migration [13]. Human β-defensins HBD-1 and HBD-3 also exhibit modulatory activity against actin regulators such as Rho family proteins, vascular endothelial growth factor (VEGF) and metastasis-associated 1 family member 2 (MTA2), resulting in anti-migratory effects against oral squamous cell carcinoma [14,15], head and neck [16] and colon cancer cells [17]. Of note, phosphoinositides, especially PI(4,5)P_2_, are particularly instrumental in initiating/maintaining cell polarity and cascading cytoskeletal signalling required for cell migration [18]. It is, therefore, of interest to determine whether PI(4,5)P_2_-binding defensins could suppress tumour cell migration, in addition to direct tumour cell lysis.

Human β-defensin 2 (HBD-2) is a potent antiviral, antibacterial and antifungal HDP, and is also capable of potentiating immune responses [19,20,21,22]. HBD-2 acts as a chemoattractant for dendritic cells, T cells and monocytes, is a chemokine receptor CCR2/CCR6-dependent manner [23,24,25,26]. Therefore, HBD-2 was proposed as an antiviral and anti-colitis agent as well as a vaccine adjuvant [19,21]. In contrast to the well-studied antimicrobial and immune-modulating properties, the effects of HBD-2 on tumour cells are still poorly characterised. Indeed, there are only a few expression studies to date, and these have simply correlated the endogenous HBD-2 level to tumour cell proliferation and invasion, suggesting opposing effects depending on the cancer setting. Therefore, based on our reported interaction between HBD-2 and PI(4,5)P_2_ [27], we aimed to assess the action of HBD-2 against tumour cells by investigating the oncolytic and anti-migratory effects of endogenous HBD-2. Here, our cell biological and confocal microscopy data reveal that HBD-2 kills tumour cells via an acute, non-apoptotic, membrane blebbing-associated cell lysis mechanism. In addition, HBD-2 does not appear to affect actin-dependent tumour cell migration, despite the reported PI(4,5)P_2_ binding. Together, our findings provide further insights into defensin biology and inform future defensin-based drug development. 

## 2. Materials and Methods

### 2.1. Expression of HBD-2 in Pichia pastoris

HBD-2 was recombinantly expressed in the methylotrophic yeast *Pichia pastoris* (GS115) and purified using SP-Sepharose cationic exchange chromatography, as previously described [28].

### 2.2. Cell Lines and Cultures

Human epithelial cervical cancer (HeLa) and leukemic monocytic lymphoma (U937) cells were cultured in RPMI-1640 (Invitrogen, Carlsbad, CA, USA) media supplemented with 5–10% (*v*/*v*) foetal calf serum (FCS), 100 U/mL penicillin and 100 μg/mL streptomycin (Invitrogen). Mouse embryonic fibroblast (MEF) and MEF Bax/Bak knockout cells were a kind gift from the Puthalakath lab at La Trobe University [29] and cultured in DMEM-F12 (Invitrogen) supplemented with 10% (*v*/*v*) FCS, 100 U/mL penicillin and 100 μg/mL streptomycin. Human breast adenocarcinoma (MDA-MB-231) and basal cell carcinoma (BCC) cells were cultured in DMEM (Invitrogen) supplemented with 10% (*v*/*v*) FCS, 100 U/mL penicillin and 100 μg/mL streptomycin. Breast adenocarcinoma (MCF-7) cells were cultured in EMEM (Sigma-Aldrich, St. Louis, MI, USA) with 10% (*v*/*v*) FCS, 100 U/mL penicillin and 100 μg/mL streptomycin. All cell lines were cultured at 37 °C in a humidified atmosphere containing 5% CO_2_.

### 2.3. Cell Viability Assay

Different concentrations of HBD-2 (0–50 µM) were added to pre-optimised cell densities for each cell line (1 × 10^5^ cells/mL) in 96-well plates in an appropriate complete medium. Following 48 h incubation with HBD-2, cell viability was determined using MTT and MTS reagents as previously described [10]. 

### 2.4. Propidium Iodide (PI) Uptake Assay

Cells at 1 × 10^6^ cells/mL in serum-free medium containing 0.1% (*w*/*v*) bovine serum albumin (BSA) (Sigma-Aldrich) were treated with varying concentrations of HBD-2 at 37 °C for 30 min, as previously described [10]. Cells were subjected to flow cytometry analysis using BD FACSCanto II Flow Cytometer and BD FACSDiva Software v8.8.10 (BD Biosciences, San Jose, CA, USA).

### 2.5. ATP Bioluminescence Assay

ATP release assay was conducted using an ATP bioluminescence assay kit (Roche Diagnostics, Mannheim, Germany). U937 and HeLa cells were suspended at 1 × 10^6^ cells/mL in PBS containing 0.1% (*w*/*v*) BSA and mixed with luciferase/luciferin reagent at a ratio of 4:5. The mixture was added to HBD-2 samples and the level of ATP release was measured immediately as bioluminescence emission signal intensity for 30 min with 30 s intervals.

### 2.6. Lipid Inhibition Assay

HBD-2 (25 µM) or PBS was incubated with 10 µM phosphatidic acid (PA), phosphatidylinositol-3,5-bisphosphate (PI(3,5)P_2_) or phosphatidylinositol-4,5-bisphosphate (PI(4,5)P_2_) (Avanti Polar Lipids, Birmingham, AL, USA) on ice for 30 min, followed by flow cytometry-based PI uptake assay with U937 cells.

### 2.7. Caspase-Glo Assay

Caspase-Glo 3/7 assay reagent (Promega, Madison, WI, USA) was used to detect caspase activation in vitro. U937 and HeLa cells were seeded at 1 × 10^5^ cells/mL, followed by treatment with HBD-2 (25 µM) for 30 min. Caspase-Glo reagent was added in 1:1 ratio, incubated in dark for 1 h at RT, followed by measuring luminescence. 

### 2.8. Confocal Laser Scanning Microscopy (CSLM)

Live imaging was performed on a Zeiss LSM-780 confocal microscope using a 63× oil immersion objective in a 37°C incubator with 5% CO_2_. Adherent HeLa cells were cultured overnight on coverslips while suspension U937 cells were immobilised onto 0.01% (*w*/*v*) poly-L-lysine-coated coverslips. Both cell types were prepared in serum-free RPMI 1640 medium containing 0.1% (*w*/*v*) BSA and 2 μg/mL PI. HBD-2 was added to directly to the imaging chamber to final concentration of 25 μM. Excitation and emission wavelengths were 488 nm and 514 nm (for green channel, PKH67), and 514 nm and 633 nm (for red channel, PI), respectively. 

Cytoskeletal microscopy was performed on a Zeiss LSM-800 confocal microscope (Zeiss, Jena, Germany) using a 63× oil immersion objective in a 37°C incubator with 5% CO_2_. Adherent MDA-MB-231 cells were cultured overnight with SiR-actin or SiR-tubulin in serum-free RPMI 1640 medium containing 0.1% (*w*/*v*) BSA. HBD-2 (5 μM), cytochalasin D (10 μM) and nocodazole (20 μM) were added directly to the imaging chamber via a capillary tube. Excitation and emission wavelengths were 514 nm and 633 nm for the red channel, respectively. Cytoskeletal microscopy was quantified using relative cell surface area on Fiji/ImageJ software (GNU General Public License). 

### 2.9. Transwell Cell Migration Assay

Serum-free MDA-MB-231 cells were seeded at 2.5 × 10^5^ cells/mL on the inserts of Corning 6.5 mm transwell with 8.0 μm pore polycarbonate transwell plates (Corning, NY, USA). Cells were incubated for 1 h at 37 °C, 5% CO_2_ prior to treatment with nocodazole (20 μM), cytochalasin D (10 μM) or HBD-2 (5 μM) in serum-free DMEM to the insert with the cells. Serum-free or serum-DMEM were added to lower wells and the cells were incubated for 6 or 24 h at 37°C, 5% CO_2_. Migrated cells were fixed with 70% ethanol for 10 min at RT, followed by staining with 0.2% crystal violet for 10 min at RT. Membrane inserts were imaged using Olympus BX41 microscope and Olympus DP25 camera, with 4× (data not shown) and 10× oil immersion objective lenses, followed by absorbance reading of the stain dissolved with 10% (*v*/*v*) acetic acid at 590 nm. Relative migration was measured by the migration index, denoted by the following equation: $$Migration\;index=\frac{Treatment-no\;FCS\;control}{FCS-no\;FCS\;control}$$

## 3. Results

### 3.1. HBD-2 Induces Tumour Lytic Cell Death, Independent of Apoptosis 

To determine the anti-tumour activity of HBD-2, tetrazolium-based cell viability assays were performed. We found that in a dose-dependent manner, HBD-2 killed various tumour cell lines, including cervical cancer cells (HeLa), prostate cancer cells (PC3), breast cancer cells (MCF-7 and BCC) with IC_50_ of ≥50 μM, with minimal cytotoxic effects against monocytic lymphoma cells (U937) and metastatic breast carcinoma cells (MDA-MB-231) (Figure 1a).

As PI(4,5)P_2_-binding defensins can induce lytic cell death, characterised by large membrane blebbing and membrane permeabilisation, we investigated the ability of HBD-2 to lyse tumour cells. First, flow cytometry analysis was conducted using propidium iodide (PI) as a membrane permeabilisation indicator. We observed moderate, dose-dependent levels of membrane permeabilisation, with PI positivity sitting at 10–30% in most tested cells lines at 50 μM HBD-2, except for MDA-MB-231 (at only ~5%) (Figure 1b). To complement the PI uptake assay and to study the kinetics of oncolysis, we also performed an ATP release bioluminescence assay on the tumour cell lines. Indeed, HBD-2-treated HeLa cells (Figure 1c) and, to a lesser extent, U937 cells (Figure 1d) significantly showed detectable bioluminescence signals in a time-dependent fashion, suggesting the leakage of cytosolic ATP due to oncolysis. To confirm the relevance of HBD-2–lipid interaction in oncolysis, flow cytometry-based PI uptake analysis was performed with HBD-2 pre-treated with PI(4,5)_2_ (implicated in defensin-induced oncolysis), PI(3,5)P_2_ (same charge as PI(4,5)_2_ but higher affinity) and PA (non-binder) [27]. Among all tested lipids, only PI(4,5)P_2_ significantly reduced HBD-2 activity on HeLa cells (Figure 1e).

To investigate a possible role for apoptosis in the anti-tumour cell activity of HBD-2, we repeated the tetrazolium-based cell viability and PI uptake assays using Bax/Bak double-knockout MEF cells. Compared to the wild-type control, the deficiency of the pro-apoptotic proteins Bax and Bak did not impair HBD-2-induced cytotoxicity and cytolysis, indicating a non-apoptotic mechanism for HBD-2 activity (Figure 1f). Consistently, unlike apoptosis-inducing BH3 mimetics, apoptotic caspases 3/7 were not activated upon HBD-2 treatment over 30 min or 48 h periods (Figure 1g,h). 

Next, we performed confocal laser scanning microscopy to visualise HBD-2-induced oncolysis. Intriguingly, HBD-2-treated HeLa and U937 cells display characteristic lytic morphologies, evident by strong PI positivity, loss of cell integrity and formation of large membrane blebs (Figure 2a). Time-course images of HBD-2 treated U937 were also captured to provide an overview of HBD-2 mediated cell permeabilisation (Figure 2b). The addition of HBD-2 rapidly led to the formation of membrane blebbing and detection of slight PI staining, indicating an early sign of membrane permeabilisation (01:00 min). A flashing expulsion of cellular content was then observed in these cells, as indicated by a flux of nucleic acid extracellularly, which was immediately stained by PI-containing medium in a few seconds. The subsequent damage to the plasma cell membrane then leads to the increased PI influx, and thus enhanced red fluorescent signal. 

### 3.2. HBD-2 Does Not Affect Cytoskeleton-Dependent Tumour Cell Migration 

Many defensins, such as PvD1, HBD-1 and HBD-3 have been shown to perturb tumour cell migration, often via the disruption of cytoskeleton dynamics [13,14,17,30]. Since PI(4,5)P_2_ and other phosphoinositides are pivotal signalling messengers orchestrating the cell migration machinery, we, therefore, sought to determine whether the PI(4,5)P_2_-binding defensin HBD-2 could suppress tumour cell migration. To this end, a transwell migration assay was conducted on human metastatic breast cancer cells MDA-MB-231, with FBS as the migratory stimulus. Unlike cytochalasin D (actin polymerisation inhibitor) and nocodazole (tubulin inhibitor) treatments, which are known to block cell migration [31], HBD-2 did not affect FCS-induced MDA-MB-231 migration over 6 h or 24 h period (Figure 3a–c). Similar results were also observed for PC3 cells (Appendix A). Furthermore, whilst cytochalasin D and nocodazole caused drastic changes in actin network and microtubules, respectively, with a significant reduction in relative cell surface area post-treatment, HBD-2 had little effect on these cytoskeleton elements even after 24 h (Figure 3d–g).

## 4. Discussion 

The multifunctionality and broad-spectrum activities against pathogens and tumour cells of many defensins and HDPs present increasing clinical interest. Their functional complexities, however, also pose certain challenges for defensin-based drug development and require a comprehensive understanding of defensin biology. HBD-2, a prominent defensin with potent immune-modulating and antimicrobial properties, has been explored as a vaccine adjuvant and anti-infective agent [19,27,32,33,34,35]. Nevertheless, unlike other defensins, its therapeutic applicability in cancer settings remains poorly studied. Based on the strong interaction of HBD-2 to membrane PI(4,5)P_2_ lipid, we aimed to investigate the effect of HBD-2 on tumour cell-related PI(4,5)P_2_-mediated processes, namely oncolysis and tumour migration. 

Our study demonstrated that HBD-2 displays moderate cytotoxic effects against various tumour cells. To this end, HBD-2 effectively triggers membrane lysis, ultimately leading to tumour cell death, which resembles necrosis rather than apoptosis. These findings are consistent with the oncolytic activity of the plant defensin NaD1 [36]. Furthermore, as similarly reported for HBD-3 [10], only pre-treatment with PI(4,5)P_2_, not the equally charged and stronger binder PI(3,5)P_2_, substantially impaired HBD-2-induced tumour cell lysis. Biophysical studies (such as X-ray crystallography) with HBD-2 and PI(4,5)P_2_ showed the intricate nature of the interaction, with two PI(4,5)P_2_ molecules bound to HBD-2. Mutations in the cationic residues of HBD-2, specifically K25 and K36, sequestered lipid binding and antifungal activity. This provides insight into the membrane destabilising nature of HBD-2-PI(4,5)P_2_ interaction, providing further support for the acute lytic nature of HBD-2 [27]. Although it remains to be defined how HBD-2 and other defensins enter tumour cells, it was demonstrated that electrostatic interaction between defensins and membrane phospholipids led to membrane perturbation, the formation of necrotic blebs and eventually cell death [5,6,18]. Together, our findings provide further evidence for a novel conserved oncolytic mechanism among defensins [10] and emphasises the importance of targeting phosphoinositide in cell death [18].

Increased levels of phosphoinositides, particularly PI(4,5)P_2_, are well-reported during tumourigenesis [37,38,39]. PI(4,5)P_2_, through its diverse effectors, is essential for tumour cell polarity, epithelial-to-mesenchymal transition, invasion and metastasis through the recruitment of focal adhesion proteins and actin polymerisation effectors [38,40,41]. PI(4,5)P_2_-producing enzymes phosphatidylinositol 5-phosphate 4-kinase (PIP4K) and phosphatidylinositol 4-phosphate 5-kinase type 1 alpha (PIP5KIα) are also overexpressed in various cancers, such as triple-negative breast cancers, HER2-positive breast cancer, advanced prostate cancer and luminal ER+ cancer [42,43,44,45,46,47,48]. In these tumour settings, PI(4,5)P_2_ generated by PIP4K and PIP5KIα act upstream of and thus promote PI3K-Akt signalling, which is crucial for cancer growth and survival [47,49,50,51]. As the apparent determinant of HBD-2-mediated oncolysis, one can speculate that the level of plasma membrane PI(4,5)P_2_ (and potentially other phosphoinositides), which is likely to vary among different tumour cells, dictates the HBD-2 potency. For example, cervical cancer HeLa cells containing ~8% phosphoinositides, compared to 2% in U937 cells [52,53], may lead to the greater HBD-2 susceptibility, as we observed in this study. In addition, multiple physical changes of plasma membranes upon tumour transformation could also have contributed to the susceptibility to defensin-induced oncolysis [54]. Due to an upregulation of negatively charged membrane components (e.g., O-glycosylated mucins [55,56] and heparan sulfate proteoglycans [57]) and the breakdown of membrane asymmetry (e.g., phosphatidylserine externalization [58,59]), tumour cells have an increased negative charge on their membranes, thus contributing to initial electrostatic interaction of defensins, such as HBD-2, to tumour cell surface. Other factors, such as increased surface area [60,61] and increased fluidity from reduced membrane cholesterol [62,63], further make tumour cells more susceptible to the activity of defensins. Nevertheless, the precise mechanisms of differential sensitivity of different tumour cells towards HBD-2 and other defensins remain to be determined.

Of note, the anti-tumour activity of HBD-2 (≥50 µM) was lower than that reported for other plant and human defensins, such as NaD1 (~2–7 µM) and HBD-3 (~10–20 µM) [10,36], as well as its own antimicrobial activities (~1–4 µM) [27,34,64]. As the net positive charge is one of the key determinants of HDP anti-tumour activity and defensin–phospholipid interaction [5,40,65,66], the lower net positive charge of HBD-2 (net charge of +6) could be responsible for it being less potent than HBD-3 (net charge of +11). Although HBD-2 and NaD1 have a similar net charge (+6), differences in binding affinity and/or charge distribution between HBD-2–PI(4,5)P_2_ and NaD1–PI(4,5)P_2_ could contribute to the differential activity of the two defensins [8,27]. In contrast, the presence of other membrane and cell wall moieties, such as glucosylceramide in fungal cells may make microbial cells more susceptible to HBD-2 [67]. 

In contrast to many other defensins such as HBD-1, HBD-3 and PvD1 [13,14,17,30], HBD-2 does not appear to affect tumour cell migration. Often, through various mechanisms, defensins perturb the cytoskeleton, a key component of the cell migration machinery, and are orchestrated by phosphoinositide signalling. However, this was not observed for HBD-2. It is tempting to speculate whilst HBD-2 can bind PI(4,5)P_2_, the HBD-2–PI(4,5)P_2_ binding interaction may be competed for by other PI(4,5)P_2_-binding cytoskeletal proteins. Interestingly, HBD-2 has been reported to promote the chemotactic migration of immune cells and keratinocytes [23,26]. In addition to membrane phospholipids, HBD-2 may interact with other membrane targets that could mediate the migratory effect of HBD-2 on immune cells and keratinocytes. For example, HBD-2 may mediate cell migration in a PI(4,5)P_2_-independent, CCR6-regulated manner, leading to F-actin accumulation [68]. Other pathways such as EGFR/STAT3 [23], VEGF signalling [24] and GPCR/ERK/JNK/p38 [25] may also play crucial roles in HBD-2 mediated cell migration. Thus, the binding of HBD-2 to PI(4,5)P_2_ may not be the only determinant, as cell migration can occur in a PI(4,5)P_2_-independent manner.

Overall, the low mammalian cell cytotoxic nature of HBD-2, along with its potent activity towards microbial cells, reassures the potential of HBD-2 as a less-toxic anti-infective therapeutics. The exploitation of HBD-2 for its potent antimicrobial and immunomodulatory activity in disease settings, such as experimental colitis, asthma and sepsis, demonstrates promising results, with the abrogation of disease, with minimal side effects in vivo [21,22,69,70]. Our data from PI uptake assay, ATP release assay, confocal microscopy and caspase activity assay consistently show that at high concentration, HBD-2 can rapidly permeabilise tumour cells and induce cell lysis, but not apoptotic cell death. The discrepancy between MTT and PI uptake/ATP release assay data may be attributed to several factors, particularly the presence of serum in the MTT assay, which is a major extrinsic factor impairing HBD-2 activity [71]. The presence of serum and serum proteases, especially during prolonged periods (48 h), may result in protein degradation, hence affecting the cytotoxic activity of HBD-2. Indeed, HBD-2 loses its antibacterial activity in the presence of 20% serum [33]. It is, therefore, reasonable to speculate that serum presence in MTT assay may also reduce the anticancer activity of HBD-2. However, whether HBD-2 should be explored for anticancer therapy, the modest anti-tumour effects of HBD-2 shown in our study suggests an effective peptide delivery and/or substantial peptide engineering would likely be required. Indeed, HBD-2 transfection induces oral carcinoma cell death and impaired tumour invasion [72], suggesting that HBD-2 may work more potently once delivered directly into tumour cells. In addition, a recombinantly engineered peptide derived from HBD-2 and oncolytic vaccinia virus significantly enhances anti-tumour immune response, inhibiting tumour growth [73], emphasising the importance of engineered HBD-2 for enhanced activity. 

## 5. Conclusions

Together, our findings offer further insights into HBD-2 biology and support the concept of a non-apoptotic, oncolytic mechanism conserved in defensins. In addition, it also suggests that HBD-2 does not influence tumour cell migration in vitro. These findings on the anti-tumour cell effects of HBD-2 will help inform future defensin-based drug development.

## Figures and Tables

**Figure 1 biomolecules-12-00264-f001:**
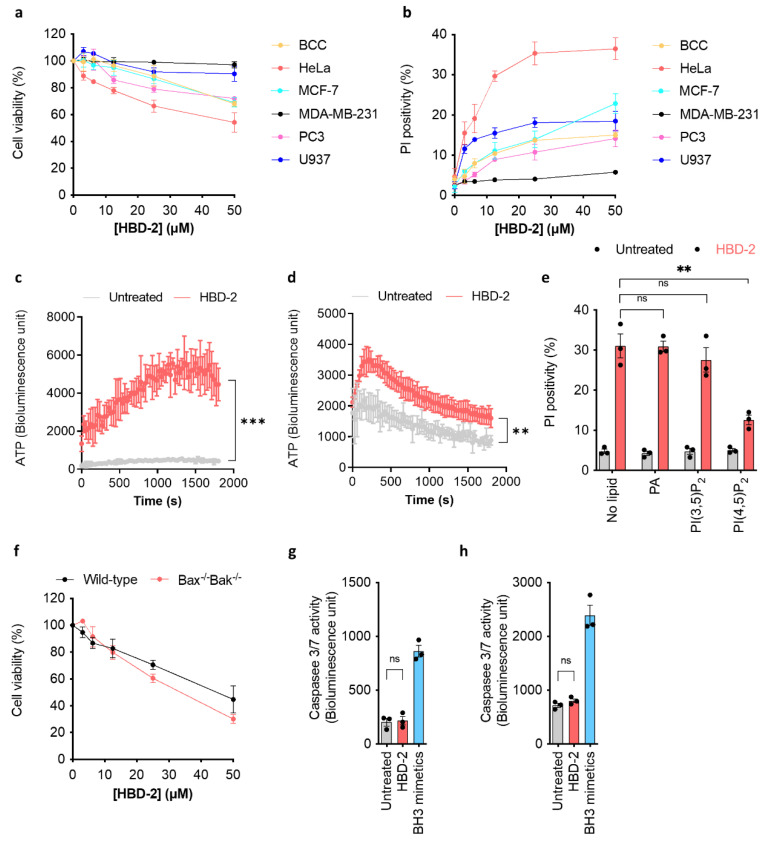
HBD-2 induces tumour cell death in a non-apoptotic manner. (**a**) Dose-dependent reduction in cell viability of different tumour cell lines (BCC, HeLa, MCF-7, MDA-MB-231, PC3 and U937) determined by tetrazolium-based assay. Data normalised against untreated control, which was arbitrarily assigned 100% viability. (**b**) Flow cytometry-based PI uptake assay on tumourigenic BCC, HeLa, MCF-7, MDA-MB-231, PC3 and U937 cells. The level of permeabilisation was expressed as PI positivity. ATP bioluminescence assay of HeLa (**c**) and U937 (**d**) cells treated with HBD-2 (50 µM) titrations. The level of ATP released was detected as bioluminescence emission signal intensity. Data represent mean ± SEM of three independent experiments. ** *p* < 0.01, *** *p* < 0.001; Two-way ANOVA. (**e**) Flow cytometry-based PI uptake assay on HBD-2 or PBS pre-incubated with PA, PI(3,5)P_2_ or PI(4,5)P_2_ lipids on U937 cells. ** *p* < 0.01, ns: not significant; unpaired *t*-test. (**f**) Dose-dependent reduction in cell viability of MEF wild-type and MEF Bax/Bak double knockout cells. Caspase-Glo 3/7 activity assay on U937 (**g**) and HeLa (**h**) cells treated with HBD-2 or BH3 mimetics over 30 min. Data represent the mean ± SEM of three independent experiments. ns: not significant; unpaired *t*-test.

**Figure 2 biomolecules-12-00264-f002:**
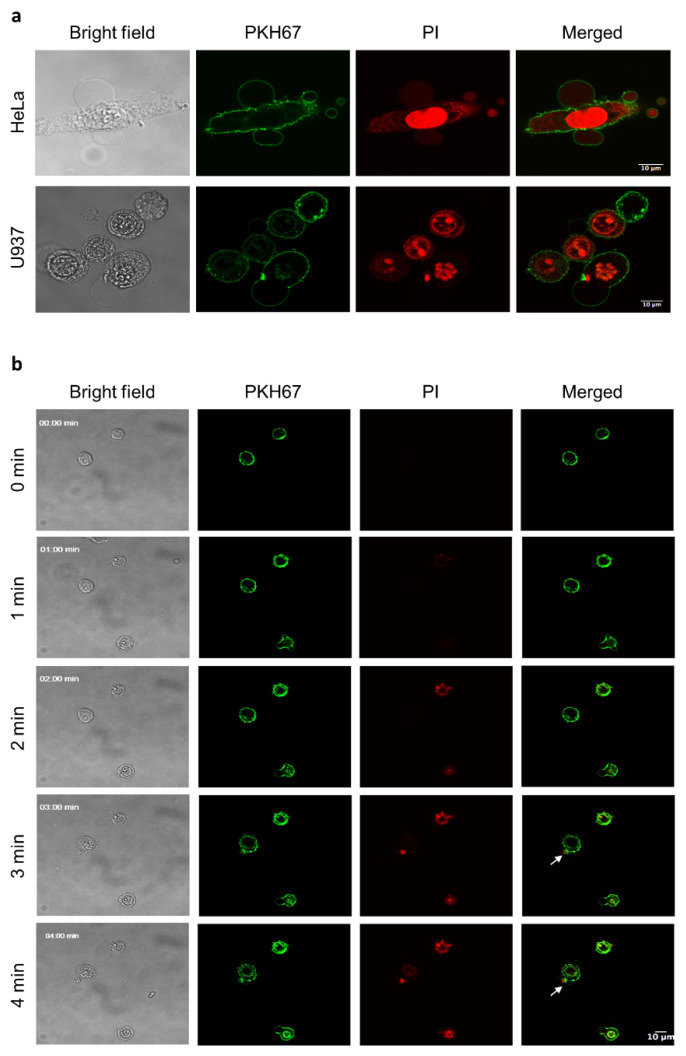
Kinetics of HBD-2-induced tumour cell death. (**a**) CLSM of HBD-2 (25 μM) treated HeLa and U937 cells, in the presence of PI and membrane stain PKH67. Images were taken 5 min post addition of HBD-2. (**b**) CLSM of HBD-2 (25 μM) on the kinetics of U937 cell permeabilisation. PI-positive staining and membrane blebbing occur 1 min post-HBD-2 addition, followed by the release of intracellular content (indicated by white arrows). Data representatives of three independent experiments. Scale bars represent 10 μm.

**Figure 3 biomolecules-12-00264-f003:**
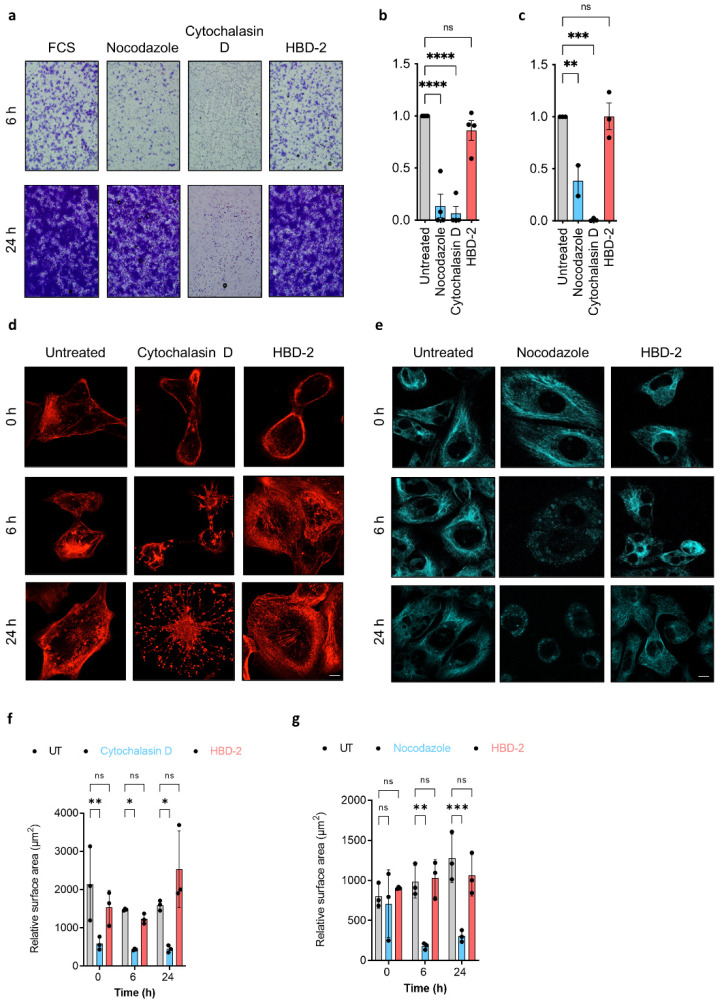
Effect of HBD-2 on tumour cell migration. (**a**) Microscopic images of MDA-MB-231 cells taken over 6 and 24 h representing migrated cells (purple) following various treatments (FCS, nocodazole, cytochalasin D and HBD-2). Quantitative analysis of membranes from **a.**, over 6 h (**b**) and 24 h (**c**), with all data normalised to No FCS control. Data represent mean ± SEM of at least three independent experiments. ** *p* < 0.01, *** *p* < 0.001, **** *p* < 0.0001, ns: not significant; One-way ANOVA. CLSM of MDA-MB-231 cells stained with SiR-Actin (**d**) and SiR-tubulin (**e**) and imaged over 24 h with HBD-2 (5 μM), cytochalasin D (10 μM) and nocodazole (20 μM). Quantification of CLSM with a relative surface area of actin (**f**) and tubulin (**g**) Data represents mean ± SEM of at least three independent experiments. * *p* < 0.05, ** *p* < 0.01, *** *p* < 0.001, ns: not significant; Two-way ANOVA. Scale bars represent 10 μm.

## Data Availability

The data supporting this research is available upon request.

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
