# Peer review of "Human β-Defensin 2 (HBD-2) Displays Oncolytic Activity but Does Not Affect Tumour Cell Migration"

_biomolecules, 2022, doi:10.3390/biom12020264_

Round 1
Reviewer 1 Report
The manuscript presents a study potentially useful for future considerations of HBD-2 peptide as a drug vehicle or a lead for oncolytic agent. To strengthen the conclusions and improve rigor, I would ask for the following:
- The ATP release assay presented in Fig. 1 was apparently performed as two independent assays. "Three makes a charm" - please add a third experiment.
- For the same assay, for U937 cells - please explore and comment if the ATP release is statically different between treated and untreated cells. Please comment on the early spike of ATP release in treated cells.
- Fig. 3 - please add a third experiment wherever there were just two, and some quantification (for example - signal intensity per cell).
Reviewer 2 Report
The authors demonstrated that human β-defensin 2 (HBD-2) induced acute lytic cell death rather than apoptosis. In addition, HBD-2 did not affect the actin-dependent tumor cell migration. The study is very interesting, however, several points as indicated below need to be addressed by authors to improve the quality of the article.
Major comments
1) In cell viability testing, the authors showed the anti-tumor activity of HBD-2 in dose-dependent manner, but the actual value of the effect is very low even in Hela for 48h incubation. Other cell lines (e. g. U937, MBA-MB-231) did not much killed with 50 μM HBD-2. The authors have already stated that in discussion, but should discuss intensively the reasons about that, because the authors mentioned that the oncolytic effect is the main point in the study. In addition, how about the cytotoxic effect with much higher concentration HBD-2?
2) Why the authors used the U937 cells in ATP release assay, Caspase assay, etc, and MDA-MB-231 in migration assay? HBD-2 did not showed enough cytotoxicity to those cell lines. If this is the case, the effect of HBD-2 that authors stated has no significance. The authors should perform those assays for ATP, caspase, migration, etc, by using cell lines to which the HBD-2 can elicit the cytotoxic activity.
Minor comments
The authors should check the spelling (e.g. page 2, line 80, RPMI-1640).
Round 2
Reviewer 2 Report
The authors adequately provided information and revised the manuscript.
The manuscript will be suitable for publication.